# PROTEIN-SE(3): UNIFIED FRAMEWORK AND COMPREHENSIVE BENCHMARK FOR SE(3)-BASED PROTEIN STRUCTURE DESIGN

## ABSTRACT

SE(3)-based generative models have shown great promise in protein geometry modeling and effective structure design. However, the field currently lacks a pipeline to support consistent re-training and fair comparison across different methods. In this paper, we propose Protein-SE(3), a unified framework accompanied by the comprehensive benchmark for SE(3)-based protein design. Protein-SE(3) integrates recent advanced methods, supports diverse evaluation metrics and also develops a mathematical decoupling toolkit. Specifically, recent advanced generative models designed for typical protein design tasks (unconditional generation and motif scaffolding), from multiple perspectives like DDPM (Genie1 and Genie2), Score Matching (FrameDiff and RfDiffusion) and Flow Matching (FoldFlow and FrameFlow) are systematically incorporated into our framework. All methods are re-trained on identical datasets and evaluated with consistent metrics, ensuring fair and reproducible comparison. Furthermore, the proposed decoupling toolkit abstracts the mathematical foundations of generative models, facilitating rapid prototyping of future algorithms without reliance on explicit protein structures. Taken together, our work establishes a standardized foundation for the advancing research field of SE(3)-based protein design.

## 1 INTRODUCTION

The design of protein structures is a fundamental challenge in computational biology, with far-reaching implications like drug discovery and enzyme engineering (Quijano-Rubio et al., 2020; Yang et al., 2025; Arunachalam et al., 2021). Recent advances in AI-driven methods (Jumper et al., 2021; Abramson et al., 2024; Watson et al., 2023; Lin et al., 2023) have revolutionized this field, enabling the *de novo* generation of complex, functional proteins. By operating residues in the special Euclidean group $SE(3) = \mathbb{R}^3 \rtimes SO(3)$ and respecting equivariance to rotation and translation, SE(3)-based models demonstrate remarkable quality and diversity in generating protein structures. From multiple perspectives of the diffusion process, researchers have proposed various generative models (DDPM-based (Lin & AlQuraishi, 2023; Lin et al., 2024), Score Matching-based (Yim et al., 2023; Watson et al., 2023) and Flow Matching-based (Bose et al., 2024; Yim et al., 2024) models) to design protein structures. However, due to differences in their dataset and training protocols, it remains challenging to make a fair cross-comparison of these methods. Existing benchmarks like ProteinBench (Ye et al., 2024) and Scaffold-Lab (Zheng et al., 2024) primarily focus on the inference performance, while overlooking the consistent re-training and fair comparison. Furthermore, the implementation of diffusion processes is closely tied to the processing of specific protein data, which hinders the understanding and further development of the underlying mathematical principles. All these challenges motivate us to establish Protein-SE(3), a unified training framework accompanied with comprehensive benchmark for SE(3)-based protein design.

Backend by Pytorch Lightning (Falcon et al., 2019), Protein-SE(3) is the first framework to systematically align diverse methods under identical datasets and training protocols, thereby laying a rigorous foundation for fair, apples-to-apples comparisons. Diverse evaluation metric as Quality (scTM, scRMSD), Diversity (Pairwise TM), and Novelty (Max. TM Score to PDB), are also integrated to analyze the strengths and limitations of different methods on typical protein structure design tasks.

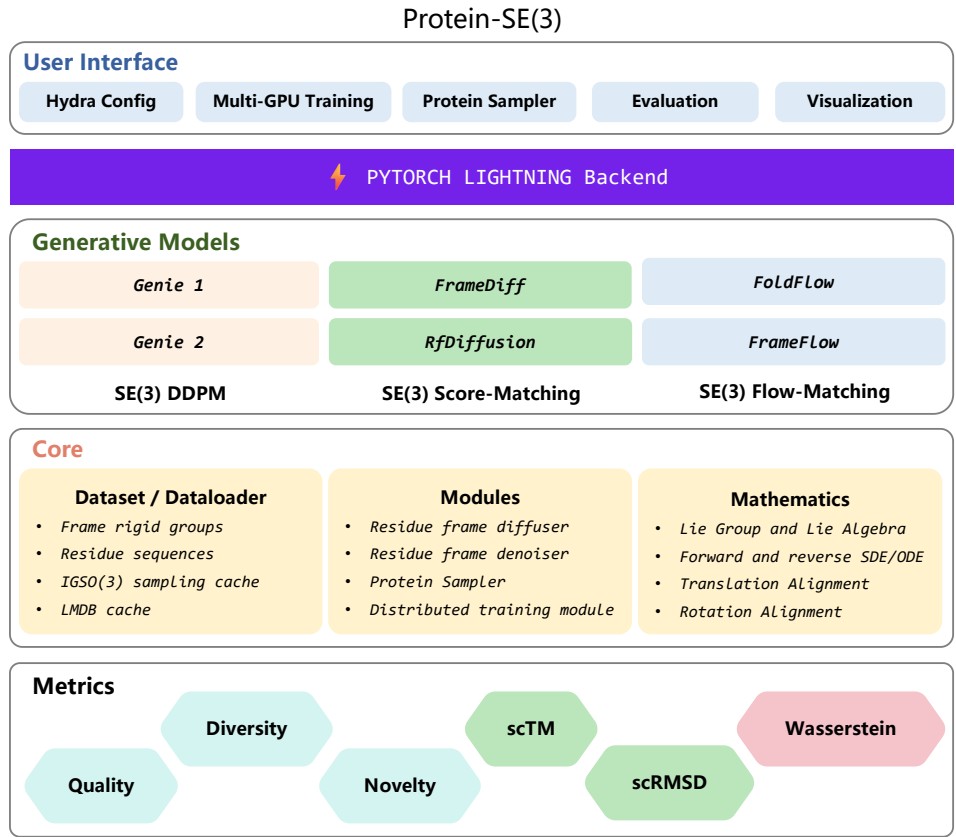

Figure 1: Overview of our proposed Protein-SE(3) benchmark.

In addition to the unified training framework and comprehensive, Protein-SE(3) also abstracts high-level mathematical principles from protein generation models into different perspectives (DDPM, Score Matching and Flow Matching). Based on the and wasserstein distance, it enables visualization and analysis of the two marginal diffusion processes in $\mathbb{R}^3$ and SO(3) spaces, facilitating agile prototyping of future algorithms without requiring explicit protein structure data. Taken together, our work is intended to equip researchers with fresh insights and foster further advances in the field. The key contributions of this paper can be summarized as follows:

- **Unified Training Framework**: We incorporate advanced SE(3)-based protein structure design methods into a unified training framework, which supports consistent re-training on identical datasets and protocols, and therefore enables fair comparisons.

- **Comprehensive Benchmark**: Diverse evaluation metrics, including Quality (scTM, scRMSD), Diversity (Pairwise TM), and Novelty (maximum TM score to PDB), are incorporated to comprehensively benchmark different methods on typical protein structure design tasks.

- **Toolkit for Mathematical Decoupling**: We present a decoupling toolkit that abstracts the mathematical principles of SE(3)-based protein design methods (core formulations are summarized in the Appendix). Through perspectives of DDPM, Score Matching, and Flow Matching, intuitive demos illustrate the distribution alignment process in both $\mathbb{R}^3$ and SO(3) spaces. This facilitates rapid prototyping and algorithm development without reliance on explicit protein structure data.

## 2 PRELIMINARIES AND PROBLEM FORMULATION

**Backbone parameterization** As shown in Figures 2(a) and 2(b), the parameterization of the protein backbone mainly follows two paths: the Alphafold2 frame (Jumper et al., 2021) and the Frenet-Serret frame (Hu et al., 2011b; Chowdhury et al., 2022). Generally, each residue is associated

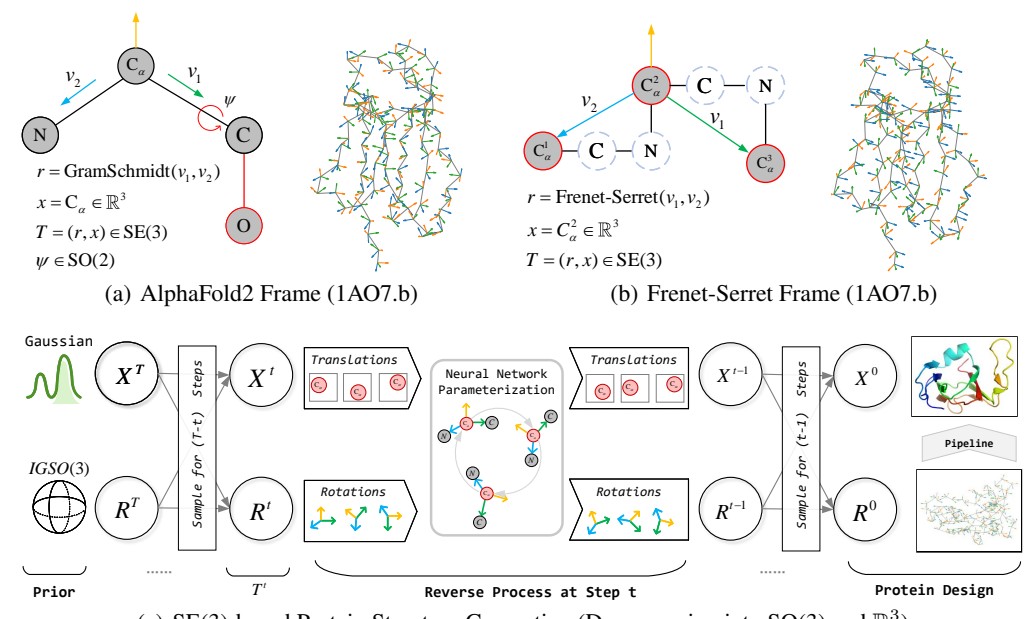

(a) AlphaFold2 Frame (1AO7.b)

(b) Frenet-Serret Frame (1AO7.b)

(c) SE(3)-based Protein Structure Generation (Decomposing into SO(3) and $\mathbb{R}^3$)

Figure 2: Protein frame parameterization and problem formulation.

with a frame, resulting in $N$ frames that are SE(3)-equivariant for a protein of length $N$: (1) In the seminal work of AlphaFold2, each frame maps a rigid transformation starting from idealized coordinates of four heavy atoms $[N^*, C_\alpha^*, C^*, O^*] \in \mathbb{R}^3$, with $C_\alpha^* = (0, 0, 0)$ being centered at the origin. Thus, residue $i \in [1, N]$ is represented as an action $T^i = (r^i, x^i) \in$ SE(3) applied to the idealized frame $[N, C_\alpha, C, O]^i = T^i \circ [N^*, C_\alpha^*, C^*, O^*]$. The coordinate of backbone oxygen atom O is constructed with an additional rotation angel $\varphi$; (2) Another way of backbone parameterization is the Frenet-Serret (FS) frame, which maps each three consecutive $C_\alpha$ into a FS frame. Following (Lin & AlQuraishi, 2023; Lin et al., 2024; Hu et al., 2011a), the FS frame $T^i$ is constructed as:

$$t^i = \frac{x^{i+1} - x^i}{||x^{i+1} - x^i||}, b^i = \frac{t^{i-1} \times t^i}{||t^{i-1} \times t^i||}, n^i = b^i \times t^i; \quad r^i = [t^i, b^i, n^i], T^i = (r^i, x^i) \quad (1)$$

where the coordinate of the second element is recognized as the translation vector $x^i$.

**Decomposing SE(3) into SO(3) and $\mathbb{R}^3$** To construct the probability path of SE(3), the definitions of an inner product and a metric on SE(3) are required to obtain a Riemannian structure (Bose et al., 2024; Yim et al., 2023). The common choices in previous studies (Yim et al., 2023; Bose et al., 2024) are:

$$\text{Inner Products:} \quad \langle r, r' \rangle_{\text{SO(3)}} = tr(rr'^T)/2 \quad \text{and} \quad \langle x, x' \rangle = \sum_{i=1}^{3} x_i x_i'$$

$$\text{Metric on SE(3):} \quad \langle (r, x), (r', x') \rangle_{\text{SE(3)}} = \langle r, r' \rangle_{\text{SO(3)}} + \langle x, x' \rangle_{\mathbb{R}^3} \quad (2)$$

As shown in Figure 2(c), the probability path of SE(3) are typically decomposed into SO(3) and $\mathbb{R}^3$. The protein structure design task is to learn the reverse process from the prior distributions (Gaussian for $\mathbb{R}^3$ and IGSO(3) for SO(3)) to the target structure distributions of actual proteins (see Appendix C for the definition of IGSO(3)).

## 3 DATASETS FOR RE-TRAINING

**Unconditional Scaffolding** Following the filtering protocol of FrameDiff(Yim et al., 2023), we construct a subset of the Protein Data Bank (PDB) (Berman et al., 2000) by selecting monomeric proteins with sequence lengths ranging from 60 to 512 residues and a structural resolution $< 5\text{Å}$. After filtering out proteins with more than 50% loops, the dataset is left with 19,703 proteins.

**Motif Scaffolding**    Motif scaffolding problems consist of sequence and structure constraints on motif(s). We use the same protein subset mentioned above but with lengths range from 60 to 320 (17,576 proteins in total). The motif masks are randomly generated during training, following the mask sampling method described in Genie2 (Lin et al., 2024). For evaluation, we use a previously published motif scaffolding benchmark Design24 (Watson et al., 2023) comprising 24 tasks curated from recent publications.

Datasets for training and evaluation are all publicly available at Harvard Dataverse (organized as the LMDB cache for efficient and parallel data loading).

## 4    BASELINE MODELS

We ensemble recent open source protein structure design baselines (Lin & AlQuraishi, 2023; Lin et al., 2024; Yim et al., 2023; Bose et al., 2024; Yim et al., 2024) in the unified training framework. The renowned method RfDiffusion (Watson et al., 2023) has received widespread recognition for excelling in *de novo* design, but its training code is unfortunately unavailable, so we report its performance with the official checkpoint as a supplementary reference.

**DDPM Methods**    combine aspects of the SE(3)-equivariant reasoning machinery of IPA with denoising diffusion probabilistic models to create a diffusion process (conditional or unconditional) over protein structures, including Genie1 (Lin & AlQuraishi, 2023) and Genie2 (Lin et al., 2024).

**Score-Matching Methods**    define a forward noising process as Brownian motion on SE(3), where translations in $\mathbb{R}^3$ and rotations in SO(3) are treated separately but consistently. The score function is then learned to estimate gradients of log-densities on this manifold, enabling efficient reverse sampling that generates realistic protein backbones or scaffolds. Representative approaches include FrameDiff (Yim et al., 2023) and RfDiffusion (Watson et al., 2023).

**Flow-Matching Methods**    is a family of continuous normalizing flow models tailored for distributions on $\text{SE}(3)^N$, which directly regress time-dependent vector fields that generate probability paths. Representative methods are FoldFlow (Bose et al., 2024) and FrameFlow (Yim et al., 2024).

## 5    METRICS

In this section, we briefly introduce the evaluation metrics that will be used to investigate different protein design methods. After re-training on identical training datasets and protocols, all methods integrated in Protein-SE(3) are fairly compared.

**Quality.**    To test whether a model generates designable proteins, we use an *in silico* self-consistency pipeline. The generated structure is first input into an inverse folding model (ProteinMPNN (Dauparas et al., 2022)) to produce 8 candidate sequences, and their structures are then predicted with ESM-Fold (Lin et al., 2023). The consistency between the generated and predicted structure is evaluated using metrics scTM and scRMSD (Zhang & Skolnick, 2005).

**Diversity.**    This metric measures the diversity of generated structures, ensuring the method produces varied backbones rather than replicating known folds. We use the average pairwise TM-score across designable samples and varying lengths as the diversity metric (lower is better).

**Novelty.**    This metric evaluates a method's ability to explore novel structural space. Novelty is measured by the maximum TM-score between each designed structure and all reference PDB structures, computed with Foldseek (Van Kempen et al., 2024) (lower is better).

**Secondary Structure Distribution.**    For generated backbones, helix and strand percentages reflect secondary structure distribution (Pelton & McLean, 2000; Venyaminov & Yang, 1996). A reasonable distribution should resemble that of natural proteins, rather than being biased toward helices or sheets (Lin et al., 2024; Yue et al., 2025).

# 6 COMPREHENSIVE BENCHMARK

In this section, we present the benchmark developed to systematically evaluate the design performance of the baseline methods. The experiments are organized as follows:

- **Benchmarking unconditional protein structure design across varying lengths.** We benchmark models for unconditional protein structure design with the aforementioned metrics including Quality, Diversity and Novelty. With length-based performance analysis, this benchmark could serve as a standardized assessment for future work.

- **Benchmarking motif scaffolding with Design24.** Motif scaffolding problems consist of structure and sequence constraints on motifs, along with length constraints on target scaffolds. We evaluate baseline models (RfDiffusion, Genie2, Frameflow) on previously published benchmark Design24 (Watson et al., 2023) with common metrics (scTM and MotifRMSD).

- **In-distribution analysis on secondary structure.** Given the protein backbones generated with different baselines, we record the percentages of the helix and strand, respectively, and visualize the distribution with respect to the percentages. In-distribution analysis assesses how well the model generalizes to natural proteins, rather than biased to a specific structural class.

- **Efficiency Analysis.** To measure the GPU memory and time needed to train an SE(3)-based generative model or generate protein structures, we report training time, inference time, step count, and model size across different methods. Although efficiency may be less critical compared to other metrics, it remains a useful metric to assess the model scalability and practicality.

## 6.1 LENGTH-BASED PERFORMANCE ANALYSIS FOR UNCONDITIONAL SCAFFOLDING

Table 1: Performance of generative models for unconditional structure design across varying lengths. We highlight the **best performance** in bold and the second-best with the underline. RfDiffusion is evaluated with the official checkpoint (since the training code is not available). The superscript ∗ indicates that the generation quality does not meet the standard (scTM ¿ 0.5), so Novelty and Diversity are excluded from the comparison.

| | **Length 100** | | | | **Length 200** | | | |
| | Quality | | Novelty | Diversity | Quality | | Novelty | Diversity |
| **Method** | scTM ↑ | scRMSD ↓ | Max TM ↓ | pairwise TM ↓ | scTM ↑ | scRMSD ↓ | Max TM ↓ | pairwise TM ↓ |
|---|---|---|---|---|---|---|---|---|
| Genie1 | 0.89±0.11 | 1.25±0.98 | 0.30±0.09 | 0.35±0.06 | 0.72±0.23 | 5.27±4.60 | 0.23±0.12 | 0.32±0.04 |
| Genie2 | 0.91±0.08 | 1.04±0.64 | **0.29±0.07** | 0.39±0.05 | 0.77±0.19 | 4.01±3.48 | 0.21±0.09 | 0.33±0.03 |
| FrameDiff | **0.92±0.04** | **0.93±0.39** | 0.39±0.13 | 0.37±0.06 | 0.81±0.16 | 3.11±3.08 | 0.34±0.13 | 0.35±0.07 |
| RfDiffusion | 0.97±0.01 | 0.52±0.10 | 0.34±0.13 | 0.39±0.07 | 0.97±0.02 | 0.63±0.15 | 0.31±0.10 | 0.35±0.06 |
| FrameFlow | 0.90±0.10 | 1.13±1.03 | 0.38±0.14 | **0.33±0.07** | **0.94±0.04** | **1.24±0.43** | 0.28±0.13 | **0.30±0.04** |
| FoldFlow-Base | 0.92±0.05 | 0.99±0.36 | 0.36±0.12 | 0.46±0.06 | 0.91±0.04 | 1.50±0.49 | 0.21±0.05 | 0.34±0.04 |
| FoldFlow-OT | 0.91±0.07 | 1.17±0.88 | 0.34±0.08 | 0.43±0.06 | 0.91±0.03 | 1.63±0.60 | **0.20±0.04** | 0.34±0.05 |
| FoldFlow-SFM | 0.87±0.06 | 1.39±0.55 | 0.33±0.10 | 0.44±0.06 | 0.82±0.22 | 3.76±3.12 | 0.21±0.05 | 0.35±0.04 |

| | **Length 300** | | | | **Length 500** | | | |
| | Quality | | Novelty | Diversity | Quality | | Novelty | Diversity |
| **Method** | scTM ↑ | scRMSD ↓ | Max TM ↓ | pair TM ↓ | scTM ↑ | scRMSD ↓ | Max TM ↓ | pair TM ↓ |
|---|---|---|---|---|---|---|---|---|
| Genie1 | 0.68±0.19 | 6.57±3.80 | 0.37±0.22 | 0.33±0.07 | 0.47±0.12* | 14.39±4.22 | 0.11±0.03 | 0.31±0.04 |
| Genie2 | 0.64±0.20 | 7.56±4.33 | **0.15±0.04** | 0.35±0.03 | 0.46±0.08* | 14.28±2.92 | 0.13±0.04 | 0.35±0.04 |
| FrameDiff | 0.72±0.13 | 5.40±2.97 | 0.34±0.15 | 0.36±0.09 | **0.64±0.19** | **9.68±5.17** | 0.20±0.08 | 0.34±0.05 |
| RfDiffusion | 0.94±0.04 | 1.04±0.89 | 0.32±0.12 | 0.38±0.04 | 0.90±0.11 | 3.65±2.95 | 0.24±0.08 | 0.37±0.06 |
| FrameFlow | **0.90±0.06** | **2.00±0.89** | 0.30±0.12 | 0.32±0.09 | 0.56±0.20 | 11.10±6.12 | 0.17±0.08 | 0.35±0.07 |
| FoldFlow-Base | 0.87±0.12 | 2.82±2.71 | 0.16±0.04 | 0.33±0.03 | 0.57±0.23 | 11.67±6.70 | **0.13±0.03** | **0.29±0.03** |
| FoldFlow-OT | 0.70±0.21 | 6.01±4.43 | 0.16±0.03 | 0.34±0.04 | 0.38±0.05* | 13.35±3.25 | 0.12±0.05 | 0.32±0.04 |
| FoldFlow-SFM | 0.68±0.23 | 7.54±6.53 | 0.17±0.04 | **0.30±0.03** | 0.37±0.08* | 15.46±3.70 | 0.12±0.03 | 0.32±0.05 |

We first retrain baseline models (except RfDiffusion), and then evaluate them for unconditional scaffolding across varying lengths. The results are presented in Table 1, which is based on our previously introduced metrics (Quality, Novelty and Diversity).

In terms of the Quality metric (scTM and scRMSD), flow-matching based methods (FrameFlow and Foldflow) demonstrates relatively better performance, which is in line with the mathematical analysis in Section 7. Novelty is also an essential metric to evaluates a method's capacity to explore new protein structures. With quality constraint (scTM>0.5), FoldFlow and Genie2 demonstrate strong performance in generating novel structures. Based on the structural diversity metric, flow-matching models still demonstrates impressive performance across the different chain lengths. It's noteworthy that the performance of all methods across various metrics shows a decline trend as the protein length increases, which suggests that these models generally struggle to create larger proteins due to the increased conformational complexity and diversity.

## 6.2 MOTIF SCAFFOLDING RESULTS ON DESIGN24

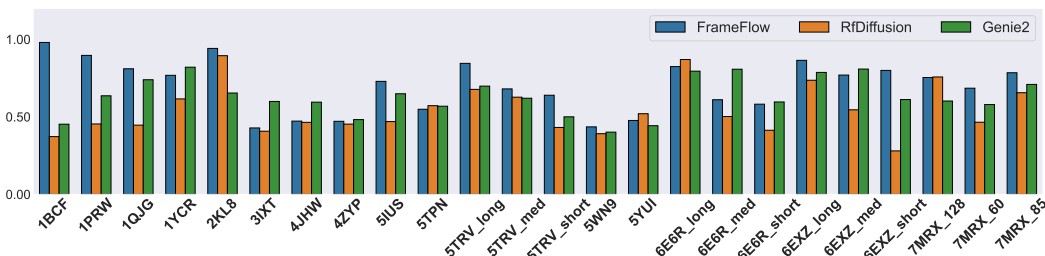

(a) Average scTM of Motif Scaffolding Designs (Higher is Better)

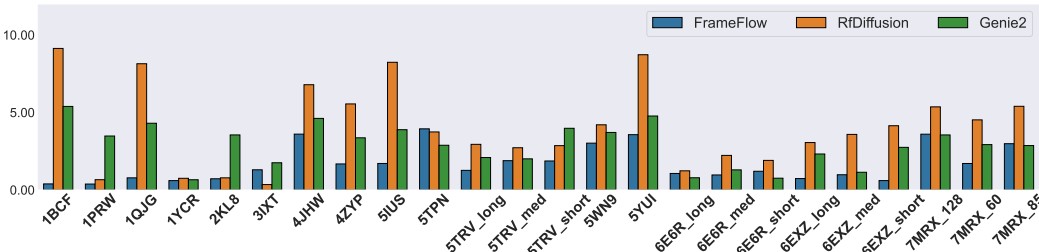

(b) Average MotifRMSD of Motif Scaffolding Designs (Lower is Better)

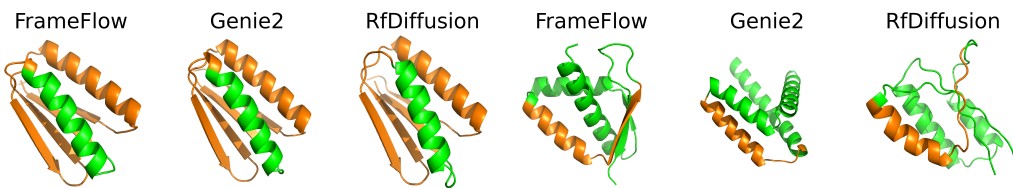

(c) Motif Scaffolding Cases (2KL8)    (d) Motif Scaffolding Cases (5TRV_med)

Figure 3: Motif scaffolding results based on Design24. In the case visualization, motif regions are colored orange and scaffolding regions green.

We consider three baselines that support the motif scaffolding task. RfDiffusion is evaluated with the official checkpoint (no source code for re-training), while FrameFlow and Genie2 are re-trained with our processed protein dataset with the length ranging from 60 to 320. with 4 scaffolding samples (using the same pipeline for quality evaluation) for each motif, the average scTM and MotifRMSD (scRSMD that only considers the motif region) of three methods are illustrated in Figure 3. Frameflow achieves the most designable scaffolds (highest scTM) among all methods in 13 out of 24 test motifs compared to Genie2's 7/24 and RfDiffusion's 6/24. For the MotifRMSD metric, FrameFlow still demonstrates superior performance by generating 19 proteins with the lowest MotifRMSD out of 24 tests, compared to the Rfdiffusion's 1/24 and Genie2's 4/24.

## 6.3 IN-DISTRIBUTION ANALYSIS ON THE SECONDARY STRUCTURE

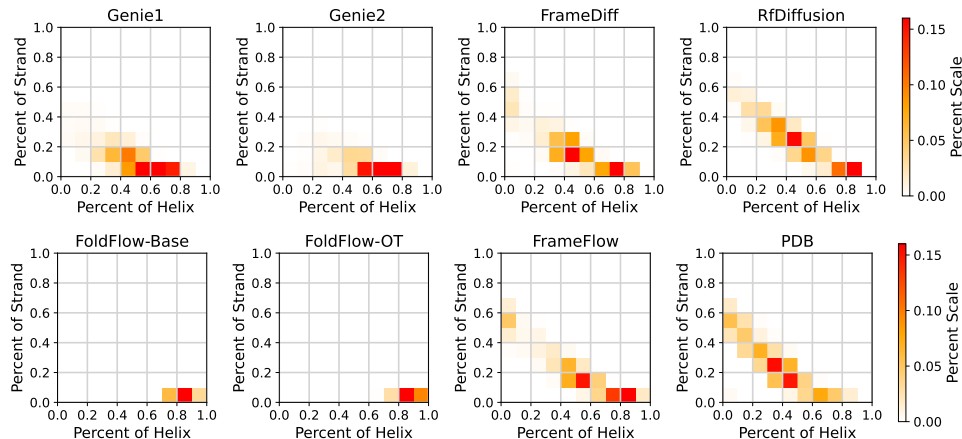

Figure 4: The distribution of secondary structure on unconditionally generated protein structures.

Given 5 unconditionally generated protein structures with every length ranging from 60 to 320 (1,040 structures in total), we report the secondary structure distribution (the percentages of helix and strand) of each method in Figure 4. The proteins generated by official RfDiffusion, re-trained FrameFlow and FrameDiff have more reasonable distributions, which is similar to those randomly sampled from the PDB (bottom right). However, the distributions of FoldFlow, Genie1 and Genie2 differ significantly from the PDB distribution, indicating the risk to generate protein backbones consistently dominated by helical structures. This explains why the performance of these methods is inferior to that of the others in the Quality and Diversity, as shown in the Table 1.

## 6.4 COMPUTATIONAL EFFICIENCY

Table 2: Efficiency comparison for protein generation models.

| Method | Epochs | Training Time | GPUs | Model Size | Sample Steps | Inference Time |
|---|---|---|---|---|---|---|
| Genie1 (Lin & AlQuraishi, 2023) | 100 | ~3.0 days | 2×A100 | 4.10M | 1k | 40 hours |
| Genie2 (Lin et al., 2024) | 100 | ~3.0 days | 4×A100 | 15.7M | 1k | 26 hours |
| FrameDiff (Yim et al., 2023) | 150 | ~4.2 days | 2×L20 | 18.8M | 100 | 31 hours |
| FoldFlow (Bose et al., 2024) | 100 | ~3.5 days | 2×L20 | 17.4M | 100 | 3.8 hours |
| FrameFlow (Yim et al., 2024) | 800 | ~2.0 days | 2×L20 | 11.3M | 100 | 1.9 hours |

To encourage developing more efficient and scalable models in the future, we also benchmark the training cost, evaluation cost, model size, and sample steps in Table 2. Note that the inference time is the total time for generating 1040 proteins with lengths ranging from 60 to 320 (following the setup for in-distribution analysis in Section 6.3). Here we found that:

- Genie1 and Genie2 consume the most GPU memory during training and time for inference, primarily due to the $O(N^3)$ scaling of triangular multiplicative update layers (Lin et al., 2024).
- As for inference efficiency, Flow-Matching methods (FrameFlow and FoldFlow) significantly outperform DDPM (Genie1 and Genie2) and Score-Matching (FrameDiff) approaches. This is primarily because flow matching leverages ordinary differential equations (ODEs) to model probability paths (Chen et al., 2023; Frans et al., 2024), allowing fewer sample steps for generation.

## 7 TOOLKIT FOR MATHEMATICAL DECOUPLING

SE(3)-based methods model and align protein structures on both the translational $\mathbb{R}^3$ and rotational SO(3) spaces. To abstract and abstract the mathematical principles underlying protein structure design

models, we developed a mathematical decoupling toolkit to visualize and analyze the distribution alignment process with the measurement of 1st-order Wasserstein distance (see Appendix C for the definition). This toolkit is built upon simple MLP layers training on the synthetic data (3D coordinates and rotation matrices represented as Euler angles), enabling systematic study on how various generative models (DDPM (Ho et al., 2020; Leach et al., 2022), Score Matching (Song et al., 2020), and Flow Matching (Bose et al., 2024)) align distributions in both the translational and rotational spaces.

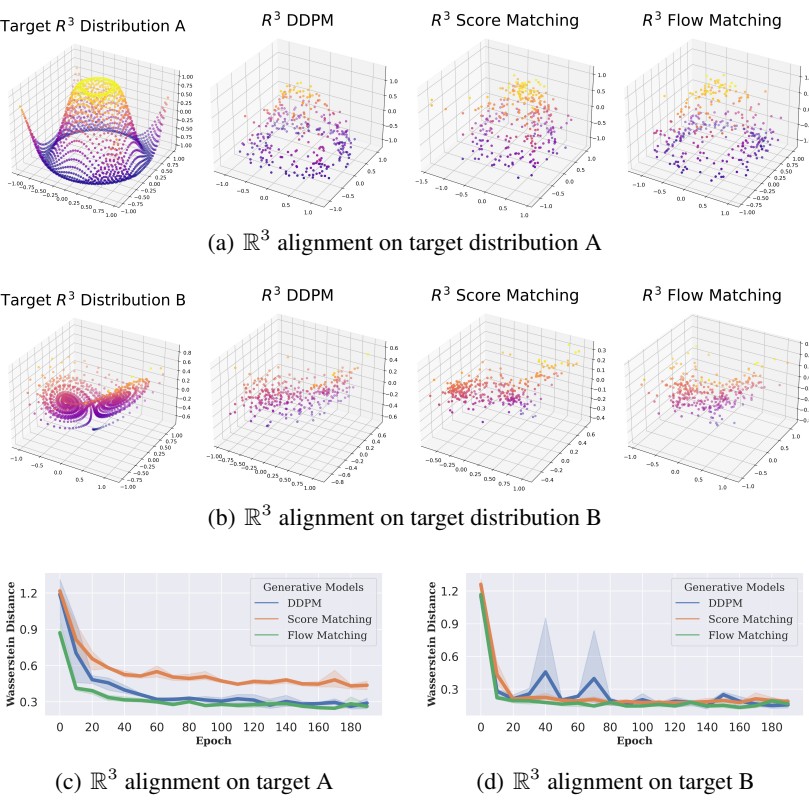

(a) $\mathbb{R}^3$ alignment on target distribution A

(b) $\mathbb{R}^3$ alignment on target distribution B

(c) $\mathbb{R}^3$ alignment on target A

(d) $\mathbb{R}^3$ alignment on target B

Figure 5: Experiments on $\mathbb{R}^3$ Alignment with different generative modeling methods.

**Translation alignment in $\mathbb{R}^3$ space** The translations ($C_\alpha$ positions) of proteins are defined in the standard $\mathbb{R}^3$ space, where the probability path can be constructed easily through the previously derived closed form equations (Luo & Hu, 2021). Detailed formulations of translation alignment, as derived from different generative models, are presented in Appendix D. While figure 5 provides a visualization of the translation alignment: as training epochs increase, the $\mathbb{R}^3$ distributions sampled by three generative models become progressively closer to the target distribution, as evidenced by the decreasing Wasserstein Distance, ultimately achieving alignment with the target distribution.

**Rotation alignment on SO(3) manifold** Different from Euclidean space, SO(3) is a Riemannian manifold (Lee, 2018). Therefore, the probability path on SO(3) has been the focus of previous studies, requiring thoughtful design for the following reasons: (1) Arithmetic operations on Riemannian manifolds are not linearly defined; (2) Modeling noise with IGSO(3) rather than a Gaussian distribution; To abstract the mathematical principles behind rotation alignment from different perspectives, we first provide definitions and formulations in Appendix E. Then we use two sets of synthetic rotation matrices as the target SO(3) distributions, which is visualized using the Euler-angle representation in Figure 6 (the range of three Euler angles is set to $[-\pi/2, \pi/2]$). With increasing training epochs, the SO(3) distributions sampled by the generative models gradually converge to the target distribution, as confirmed by the decreasing 1st-order Wasserstein Distance.

Compared to DDPM and Score Matching, the curves of Flow Matching on $\mathbb{R}^3$ and SO(3) alignment exhibit better convergence, indicating the superior design performance evaluated in Section 6.

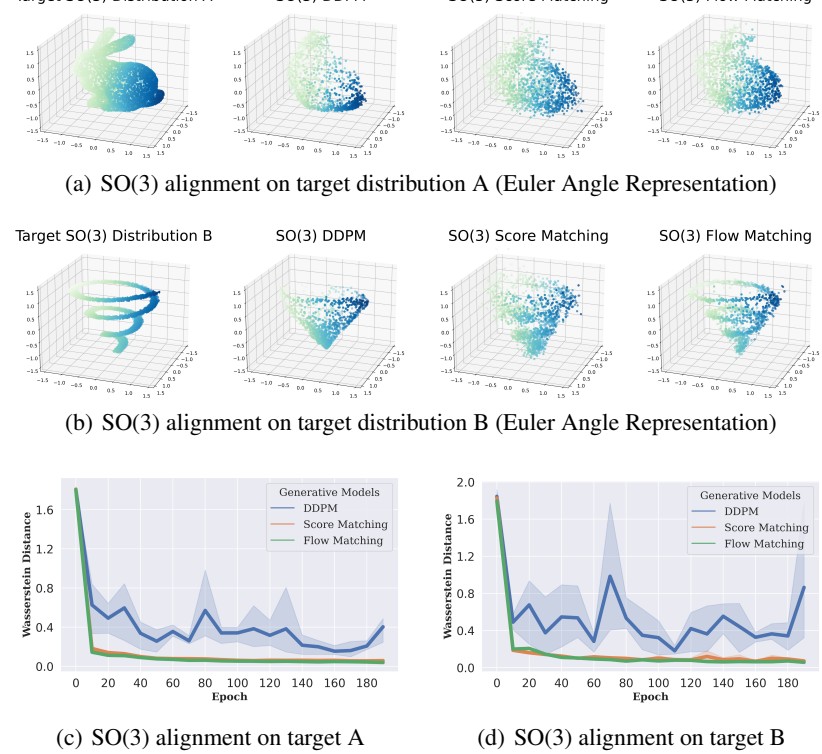

(a) SO(3) alignment on target distribution A (Euler Angle Representation)

(b) SO(3) alignment on target distribution B (Euler Angle Representation)

(c) SO(3) alignment on target A  (d) SO(3) alignment on target B

Figure 6: Experiments on SO(3) alignment with different generative modeling methods.

## 8    INSIGHTS FOR FUTURE METHOD DESIGN

To conclude our experimental results, we summarize several key insights that may inspire the design of future methods, with the aim of facilitating broader progress in the community:

- **Flow Matching as a Superior Choice:** As revealed by our mathematical decoupling toolkit, the alignment process of Flow Matching in the $\mathbb{R}^3$ and SO(3) space exhibits superior efficiency and stability. This suggests flow matching generally outperforms other models for modeling the generation process, and recent progress in the Flow Matching community (MeanFlow (Geng et al., 2025), Shortcut Model (Frans et al., 2024)) can be leveraged for future method design.

- **Distribution-level Guidance Control:** Despite most methods achieving remarkable results on certain evaluation metrics, the secondary structure distributions of their generated proteins differ substantially from that of native PDBs. This observation reveals the need to introduce distribution-level guidance into the training process (in line with ProtFID (Faltings et al., 2025)).

- **SE(3) Invariant Encoder:** Genie1 and Genie2 consume the most GPU memory during training and the most time for inference, primarily due to the scaling of triangular multiplicative update layers (Jumper et al., 2021), which indicates the need for a more efficient SE(3)-invariant encoder.

## 9    CONCLUSION AND FUTURE WORK

We propose Protein-SE(3), a unified framework and comprehensive benchmark for SE(3)-based protein design approaches, which enables fair comparison with consistent re-training on identical datasets and protocols. The developed mathematical toolkit provide intuitive demos to interpret the distribution alignment process in the $\mathbb{R}^3$ and SO(3) space. In the future, we will gradually broaden our scope beyond SE(3)-based structure generation algorithms (Geffner et al., 2025; Campbell et al., 2024; Wang et al., 2024), and will keep Protein-SE(3) updated in accordance with the latest research (some preliminary results using the official checkpoints are presented in Appendix B)

ETHICS STATEMENT

We confirm that our research adheres to the Code of Ethics. This work does not involve human subjects, personal data, or any proprietary or sensitive information. The source of constructed datasets are publicly available and widely adopted in the community, and we have followed best practices regarding licensing, documentation, and responsible use. The methods proposed are intended for advancing scientific understanding and educational purposes, without foreseeable harmful applications. We are committed to principles of fairness, transparency, and research integrity, and we have carefully considered potential concerns related to bias, privacy, and security in our study.

REPRODUCIBILITY STATEMENT

We have made every effort to ensure the reproducibility of our work. The dataset construction process is described in detail in Section 3. Furthermore, we include the complete source code in the supplementary material, allowing others to reproduce our unified framework and comprehensive benchmark. Together, these resources are intended to facilitate independent verification of our results and foster transparent scientific progress.

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

APPENDIX

## A  USE OF LARGE LANGUAGE MODELS

We used an LLM solely to aid and polish writing. Specifically, OpenAI's ChatGPT was employed for copy-editing (grammar, fluency, and tone), minor phrasing suggestions, and occasional condensation or expansion of author-written paragraphs without altering their scientific meaning.

## B  PRELIMINARY RESULTS ON RECENT METHODS

Table 3: Preliminary Evaluation Results of Recent Methods beyond SE(3) Paradigm.

| Method | scTM ↑ | scRMSD ↓ | MaxTM ↓ | PairwiseTM ↓ | Information of Protein | Architecture |
|---|---|---|---|---|---|---|
| **Length 100** | | | | | | |
| Proteina | $0.95 \pm 0.03$ | $0.75 \pm 0.23$ | $0.48 \pm 0.19$ | $0.36 \pm 0.07$ | $C_\alpha$ Coordinates Only | Flow Matching |
| Proteina-Long | $0.88 \pm 0.06$ | $1.01 \pm 0.42$ | $0.40 \pm 0.24$ | $0.32 \pm 0.05$ | $C_\alpha$ Coordinates Only | Flow Matching |
| MultiFlow | $0.98 \pm 0.01$ | $0.51 \pm 0.09$ | $0.44 \pm 0.12$ | $0.39 \pm 0.07$ | **SE(3) Frames** + Sequences | Flow Matching |
| DPLM-2 | $0.88 \pm 0.07$ | $1.31 \pm 0.65$ | $0.84 \pm 0.16$ | $0.31 \pm 0.15$ | $C_\alpha$ Coordinates + Sequences | Language Model |
| **Length 300** | | | | | | |
| Proteina | $0.87 \pm 0.06$ | $4.39 \pm 2.38$ | $0.21 \pm 0.14$ | $0.31 \pm 0.04$ | $C_\alpha$ Coordinates Only | Flow Matching |
| Proteina-Long | $0.95 \pm 0.03$ | $1.01 \pm 0.42$ | $0.40 \pm 0.24$ | $0.32 \pm 0.05$ | $C_\alpha$ Coordinates Only | Flow Matching |
| MultiFlow | $0.98 \pm 0.04$ | $0.64 \pm 0.11$ | $0.24 \pm 0.12$ | $0.33 \pm 0.05$ | **SE(3) Frames** + Sequences | Flow Matching |
| DPLM-2 | $0.94 \pm 0.05$ | $1.76 \pm 1.63$ | $0.83 \pm 0.14$ | $0.27 \pm 0.03$ | $C_\alpha$ Coordinates + Sequences | Language Model |
| **Length 500** | | | | | | |
| Proteina | $0.41 \pm 0.08$ | $44.1 \pm 5.26$ | $0.20 \pm 0.10$ | $0.23 \pm 0.03$ | $C_\alpha$ Coordinates Only | Flow Matching |
| Proteina-Long | $0.93 \pm 0.10$ | $2.22 \pm 2.13$ | $0.32 \pm 0.18$ | $0.31 \pm 0.06$ | $C_\alpha$ Coordinates Only | Flow Matching |
| MultiFlow | $0.97 \pm 0.01$ | $1.01 \pm 0.26$ | $0.33 \pm 0.06$ | $0.17 \pm 0.05$ | **SE(3) Frames** + Sequences | Flow Matching |
| DPLM-2 | $0.84 \pm 0.12$ | $3.90 \pm 2.82$ | $0.69 \pm 0.21$ | $0.28 \pm 0.13$ | $C_\alpha$ Coordinates + Sequences | Language Model |

In addition to the methods integrated in our proposed Protein-SE(3), we also have evaluate several recent approaches beyond SE(3) paradigm, including Proteina (Geffner et al., 2025), MultiFlow (Campbell et al., 2024), and DPLM-2 (Wang et al., 2024). For these methods, we use the official checkpoints released by the authors, while noting that retraining under our standardized dataset and framework is left for future work. The corresponding results are reported in Table 3, which further indicate that the method MultiFlow incorporating SE(3) information achieves superior performance, particularly on Quality-related metrics such as scTM and scRMSD, suggesting that SE(3)-based approaches merit greater attention and exploration.

## C  DEFINITIONS

**Definition 1.** IGSO(3): The Isotropic Gaussian Distribution on SO(3) is a probability distribution over the 3D rotation group SO(3), which generalizes the idea of a Gaussian (normal) distribution to the non-Euclidean manifold of 3D rotations (Nikolayev & Savyolov, 1997). Detailed sampling process is described in the Appendix E.1.

**Definition 2.** Wasserstein Distance: The Wasserstein distance (Kantorovich, 1960; Villani & Villani, 2009) measures the minimum cost required to transform one probability distribution into another. For probability distributions $\mu$ and $\upsilon$ over a metric space $\chi$ (e.g. $\mathbb{R}^3$ and SO(3)), let $\Gamma(\mu, \upsilon)$ is the set of all joint distributions on $\chi \times \chi$, the 1st-order Wasserstein distance is defined as:

$$W_1(\mu, \upsilon) = \inf_{\gamma \in \Gamma(\mu, \upsilon)} \mathbb{E}_{(x,y) \sim \gamma}[\|x - y\|] \tag{3}$$

where $\gamma(x, y)$ represents a transport plan and $\|x - y\|$ denotes the transport cost from point $x$ to $y$.

# D   THEORETICAL FORMULATIONS OF $\mathbb{R}^3$ ALIGNMENT

The translations ($C_\alpha$ positions) of proteins are defined in the standard $\mathbb{R}^3$ Euclidean space, where the probability path can be constructed easily through the previously derived closed form equations. From different perspectives (DDPM (Lin & AlQuraishi, 2023), Score Matching (Yim et al., 2023) and Flow Matching (Bose et al., 2024)), we summarize core formulations for $\mathbb{R}^3$ distribution alignment implemented in Section 7.

## D.1   TRANSLATION ALIGNMENT BASED ON DDPM

Here, we present the formulations of DDPM-based translation alignment following the description of Genie1 (Lin & AlQuraishi, 2023). Let $\mathbf{x} = [\mathbf{x}_1, \mathbf{x}_2, ..., \mathbf{x}_N]$ denote a sequence of $C_\alpha$ coordinates of length $N$. Given a sample $\mathbf{x}_0$ from the target distribution over the synthetic data, the forward process iteratively adds Gaussian noise to the sample with a cosine variance schedule $\beta = [\beta_1, \beta_2, ..., \beta_T]$, where the diffusion steps $T$ is set to 1,000:

$$q(\mathbf{x}_t \mid \mathbf{x}_{t-1}) = \mathcal{N}(\mathbf{x}_t \mid \sqrt{1 - \beta_t}\mathbf{x}_{t-1}, \beta_t\mathbf{I}) \tag{4}$$

By applying the reparameterization trick to the forward process, we have

$$q(\mathbf{x}_t \mid \mathbf{x}_0) = \mathcal{N}(\mathbf{x}_t \mid \sqrt{\bar{\alpha}_t}\mathbf{x}_0, (1 - \bar{\alpha}_t)\mathbf{I}),$$
$$\bar{\alpha}_t = \prod_{i=1}^{t} \alpha_i \ , \ \alpha_t = 1 - \beta_t \tag{5}$$

According to the derivation of DDPM, the reverse process is modeled with a Gaussian distribution:

$$p(\mathbf{x}_{t-1} \mid \mathbf{x}_t) = \mathcal{N}(\mathbf{x}_{t-1} \mid \boldsymbol{\mu}_\theta(\mathbf{x}_t, t), \boldsymbol{\Sigma}_\theta(\mathbf{x}_t, t)\mathbf{I}),$$
$$\boldsymbol{\mu}_\theta(\mathbf{x}_t, t) = \frac{1}{\sqrt{\alpha_t}}(\mathbf{x}_t - \frac{\beta_t}{\sqrt{1 - \bar{\alpha}_t}}\boldsymbol{\epsilon}_\theta(\mathbf{x}_t, t)) \ , \ \boldsymbol{\Sigma}_\theta(\mathbf{x}_t, t) = \beta_t \tag{6}$$

This reverse process requires evaluating $\boldsymbol{\epsilon}_\theta(\mathbf{x}_t, t)$ based on MLP layers, which predict the noise added at time step $t$. The loss function is defined as:

$$L = \mathbb{E}_{t, \mathbf{x}_0, \boldsymbol{\epsilon}}\left[\sum_{i=1}^{N} \|\boldsymbol{\epsilon}_t - \boldsymbol{\epsilon}_\theta(\mathbf{x}_t, t)\|^2\right]$$
$$= \mathbb{E}_{t, \mathbf{x}_0, \boldsymbol{\epsilon}}\left[\sum_{i=1}^{N} \|\boldsymbol{\epsilon}_t - \boldsymbol{\epsilon}_\theta(\sqrt{\bar{\alpha}_t}\mathbf{x}_0 + \sqrt{1 - \bar{\alpha}_t}\boldsymbol{\epsilon}_t, t)\|^2\right] \tag{7}$$

where $\boldsymbol{\epsilon} = \left[\boldsymbol{\epsilon}^1, \boldsymbol{\epsilon}^2, \cdots, \boldsymbol{\epsilon}^N\right]$ and each $\boldsymbol{\epsilon}^i \sim \mathcal{N}(\mathbf{0}, \mathbf{I})$.

## D.2   TRANSLATION ALIGNMENT BASED ON SCORE MATCHING

Following FrameDiff (Yim et al., 2023), the process of $\mathbb{R}^3$ alignment is modeled as an Ornstein–Uhlenbeck process ( also called VP-SDE (Song et al., 2020)). Still, Let $\mathbf{x} = [\mathbf{x}_1, \mathbf{x}_2, ..., \mathbf{x}_N]$ denote a sequence of $C_\alpha$ coordinates of length $N$. Converging geometrically towards Gaussian, the VP-SDE of $\mathbf{x}$ in the $\mathbb{R}^3$ space is:

$$d\mathbf{x} = -\frac{1}{2}\beta(t)\mathbf{x}dt + \sqrt{\beta(t)}d\mathbf{w} \tag{8}$$

where $\beta(t)$ is a non-negative function of $t$ to describe the time-dependent noise schedule, and $\mathbf{w}$ is the standard Wiener process. Its analytical solution is:

$$\mathbf{x}_t = \alpha(t)\mathbf{x}_0 + \sigma(t)\mathbf{z}, \quad \mathbf{z} \sim \mathcal{N}(0, \mathbf{I})$$
$$\alpha(t) = \exp\left(-\frac{1}{2}\int_0^t \beta(s)ds\right), \sigma^2(t) = 1 - \alpha^2(t) \tag{9}$$

To train a score-based model, we want the network $s_\theta(x_t, t)$ to approximate the score function $\nabla_{\mathbf{x}_t} \log p(\mathbf{x}_t \mid \mathbf{x}_0)$. Note that the conditional distribution is Gaussian:

$$p(\mathbf{x}_t \mid \mathbf{x}_0) = \mathcal{N}\left(\alpha(t)\mathbf{x}_0, \sigma^2(t)\mathbf{I}\right), \tag{10}$$

so the formulation of the true score becomes:

$$\nabla_{\mathbf{x}_t} \log p(\mathbf{x}_t \mid \mathbf{x}_0) = -\frac{1}{\sigma^2(t)}(\mathbf{x}_t - \alpha(t)\mathbf{x}_0). \tag{11}$$

With the simple MSE loss function to train the network $s_\theta(x_t, t)$:

$$L = \|s_\theta(x_t, t) - \nabla_{\mathbf{x}_t} \log p(\mathbf{x}_t \mid \mathbf{x}_0)\|^2, \tag{12}$$

the reverse process from $\mathbf{x}_t$ to $\mathbf{x}_{t-1}$ can be formulated as:

$$d\mathbf{x} = \left[-\frac{1}{2}\beta(t)\mathbf{x} + \beta(t)s_\theta(x, t)\right] dt + \sqrt{\beta(t)}d\bar{\mathbf{w}}, \tag{13}$$

where $\bar{\mathbf{w}}$ is denotes Wiener process run backward in time.

### D.3 TRANSLATION ALIGNMENT BASED ON FLOW MATCHING

Flow Matching in the $\mathbb{R}^3$ space aims to learn a time-dependent velocity field $u_\theta(\mathbf{x}, t)$ such that the solution to the following ordinary differential equation (ODE):

$$\frac{d\mathbf{x}}{dt} = u_\theta(\mathbf{x}, t), \tag{14}$$

which maps samples from a known base distribution $p_0(\mathbf{x}_0)$ to samples from the target distribution $p_1(\mathbf{x}_1)$ over time $t \in [0, 1]$. In the $\mathbb{R}^3$ space, the target velocity at time $t$ is simply:

$$v_t = \mathbf{x}_1 - \mathbf{x}_0 \tag{15}$$

The model is trained to predict this velocity via a regression loss:

$$\mathcal{L}(\theta) = \mathbb{E}_{\mathbf{x}_0, \mathbf{x}_1, t}\left[\|u_\theta(\mathbf{x}_t, t) - (\mathbf{x}_1 - \mathbf{x}_0)\|^2\right] \tag{16}$$

Once the model $u_\theta(\mathbf{x}, t)$ is trained, it defines a velocity field over space and time. The sampling process amounts to integrating the learned ODE from a noise sample to a data sample:

$$\mathbf{x}_{t-1} = \mathbf{x}_t + u_\theta(\mathbf{x}_t, t)\, dt \tag{17}$$

Solvers like Euler method and Runge-Kutta can be further used to numerically integrate the ODE for higher speed and accuracy.

## E THEORETICAL FORMULATIONS OF SO(3) ALIGNMENT

Different from Euclidean space, SO(3) is a Riemannian manifold. Therefore, the diffusion process (or probability flow) on SO(3) has been the focus of previous studies, requiring thoughtful design. Here we provide the formulations of rotation alignment in the SO(3) space, which is implemented from different perspectives(DDPM, Score Matching and Flow Matching) in Section 7.

### E.1 PRELIMINARIES

**The exponential and logarithmic maps.** Generally speaking, the exponential and logarithmic relate the elements in Lie Group (rotation matrices) to the ones in Lie Algebra (skew-symmetric matrices). The skew-symmetric matrices in Lie Algebra can be specified with a rotation vector $\mathbf{\Phi}$:

$$\hat{\mathbf{\Phi}} = \begin{bmatrix} 0 & z & -y \\ -z & 0 & x \\ y & -x & 0 \end{bmatrix}, \text{ where } \mathbf{\Phi} = (x, y, z). \tag{18}$$

The magnitude of this vector $\omega = \|\mathbf{\Phi}\|$ represents the angle of rotation, and its direction $\mathbf{n} = \mathbf{\Phi}/\omega$ denotes the axis of rotation. Following the Rodrigues formula, the exponential map (rotation vector $\mathbf{\Phi}$ to rotation matrix $r$) can be simplified to a closed form:

$$r = \exp(\hat{\mathbf{\Phi}}) = cos(\omega)I + sin(\omega)\hat{\mathbf{n}} + (1 - cos(\omega))\mathbf{n}\mathbf{n}^T \tag{19}$$

Similarly, the matrix logarithm can be expressed using the rotation angle:

$$\hat{\mathbf{\Phi}} = \log(r) = \frac{\omega}{2sin\omega}(r - r_T), \ \omega = \arccos[(\text{Trace}(r) - 1)/2] \tag{20}$$

**Sampling from IGSO3($\mu$,$\epsilon^2$)** To form a rotation matrix $r \sim \text{IGSO3}(\mu, \epsilon^2)$, we first perform a random sampling from IGSO3($\mathbf{I}, \epsilon^2$). The axis of rotation $\mathbf{n}$ is uniformly sampled, and the rotation angle $\theta$ is given by the following CDF (Nikolayev & Savyolov, 1997):

$$f(\omega) = \sum_{\ell=0}^{\infty} (2\ell + 1) \exp\left(-l(l+1)\epsilon^2\right) \frac{\sin((\ell + 1/2)\omega)}{\sin(\omega/2)}, \tag{21}$$

which together yield a rotation vector $\mathbf{\Phi} = \omega\mathbf{n}$. Then the rotation vector is shifted by the mean of the distribution to obtain $R = \mu\exp(\hat{\mathbf{\Phi}})$ as the sampled rotation matrix.

### E.2 ROTATION ALIGNMENT BASED ON DDPM

Following the previous work (Leach et al., 2022), we implement a SO(3) DDPM model based on several MLP layers for the rotation alignment in Section 7. With definitions described in Section E.1, the rotation matrix can be scaled by converting them into the Lie algebra (rotation vector), element-wise multiplying by scalar value, and converting back to rotation matrix through exponential map. The matrix scaling operation is defined as:

$$\lambda(c, r) = \exp(c \log(r)), \tag{22}$$

where $\lambda(...)$ is the geodesic flow from $\mathbf{I}$ to $R$ by the amount $c$. Applying these to equations from the original DDPM model we arrive at the following definitions:

$$q\left(r_t \mid r_0\right) = \text{IGSO3}\left(\lambda\left(\sqrt{\bar{\alpha}_t}, r_0\right)\right), \left(1 - \bar{\alpha}_t\right));$$
$$p\left(r_{t-1} \mid r_t, r_0\right) = \text{IGSO3}\left(\tilde{\mu}\left(r_t r_0\right), \tilde{\beta}_t\right) \tag{23}$$

and

$$\tilde{\mu}\left(r_t, r_0\right) = \lambda\left(\frac{\sqrt{\bar{\alpha}_{t-1}}\beta_t}{1 - \bar{\alpha}_t}, r_0\right) \lambda\left(\frac{\sqrt{\alpha_{t-1}}\left(1 - \bar{\alpha}_{t-1}\right)}{1 - \bar{\alpha}_t}, r_t\right) \tag{24}$$

where $\beta$ and $\alpha$ are schedule values in line with the description in Section D.1. To train the DDPM model $\epsilon_\theta(R_t, t)$ that predicts $R_0$, the loss function is formulated as follows:

$$L = \mathbb{E}\|\epsilon_\theta(r_t, t)r_0^T - I\|_F^2, \tag{25}$$

where $\|\cdot\|_F$ represents Frobenius norm.

### E.3 ROTATION ALIGNMENT BASED ON SCORE MATCHING

We abstract the mathematical principles behind score matching for SO(3) from the previous work FrameDiff (Yim et al., 2023). For any $t \in [0, 1]$ and $r_0 \in \text{SO}(3)$, it assumes that $r_t \sim \text{IGSO3}(r_0, t)$, which is stated as the Brownian motion on SO(3). To train a score-based model on SO(3), we want the network $s_\theta(r_t, t)$ to approximate the score function $\nabla_{r_t} \log p\left(\mathbf{r}_t \mid \mathbf{r}_0\right)$:

$$\nabla_{r_t} \log p\left(r_t \mid r_0\right) = \frac{r_t}{r_0^\top r_t} \log\left\{r_0^\top r_t\right\} \frac{\partial_\omega f\left(\omega(r_0^\top r_t), t\right)}{f\left(\omega(r_0^\top r_t), t\right)} \tag{26}$$

where $\omega(r)$ is the rotation angle in radians for any $r \in \text{SO}(3)$. In our toolkit, we use MSE loss to train the MLP network $s_\theta(r_t, t)$:

$$L = \|s_\theta(r_t, t) - \nabla_{r_t} \log p(r_t \mid r_0)\|^2. \tag{27}$$

The reverse process from $r_t$ to $r_{t-1}$ can be formulated as:

$$r_{t-1} = r_t[g(t)^2 s_\theta(r_t, t)dt + g(t)d\bar{\mathbf{w}}], \tag{28}$$

where $g(t)$ is the diffusion coefficient and $\bar{\mathbf{w}}$ is the time-reversal Wiener process.

### E.4 ROTATION ALIGNMENT BASED ON FLOW MATCHING

According to the method described in FoldFlow (Bose et al., 2024), we implemented the SO(3) alignment process based on Flow Matching in our toolkit. Given two rotation matrices $r_0, r_1 \in$ SO(3), the geodesic interpolation index by $t$ has the following form:

$$r_t = r_0 \cdot \exp[t \cdot \log(r_0^\top r_1)] \tag{29}$$

Flow Matching over SO(3) aims to learn a time-dependent velocity field $u_\theta(\mathbf{r}, t)$ such that the solution to the following ordinary differential equation (ODE):

$$\frac{d\mathbf{r}}{dt} = u_\theta(\mathbf{r}, t) \tag{30}$$

FoldFlow takes $\log(r_t^\top r_0)$ divided by $t$ as the target velocity at $r_t$. Thus we train the velocity field (MLP layers) $u_\theta(\mathbf{r}, t)$ with the following loss function:

$$\mathcal{L}(\theta) = \mathbb{E}_{\mathbf{r}_0, \mathbf{r}_1, t} \left\| u_\theta(\mathbf{r}_t, t) - \log(r_t^\top r_0)/t \right\|_{\mathrm{SO}(3)}^2 \tag{31}$$

where the distance induced by the $\| \cdot \|_{\mathrm{SO}(3)}$ metric is given by:

$$d_{\mathrm{SO}(3)}(r_0, r_1) = \| \log(r_0^\top r1) \|_F, \tag{32}$$

With learned velocity field $u_\theta(\mathbf{r}, t)$, the reverse process can be written as:

$$r_{t-1} = r_t \cdot \exp[r_0^\top \cdot u_\theta(r_t, t)dt] \tag{33}$$

