# OpenReview forum: "Protein-SE(3): Unified Framework and Comprehensive Benchmark for SE(3)-based Protein Structure Design"
_ICLR.cc/2026/Conference — ICLR 2026 Conference Withdrawn Submission_

### Official Review · Reviewer_krsJ · 2025-10-20

**Soundness:** 3
**Presentation:** 3
**Contribution:** 3
**Rating:** 8
**Confidence:** 4

**Summary:**

The manuscript presents Protein-SE(3), a unified framework complemented by a comprehensive benchmark for evaluating SE(3)-equivariant protein design models. The framework integrates state-of-the-art methodologies, standardizes diverse evaluation metrics, and introduces a novel mathematical decoupling toolkit. A key feature is the re-training of all integrated methods on identical datasets and their evaluation using consistent metrics, which significantly promotes fairness and reproducibility in this rapidly evolving field. The mathematical toolkit aims to establish a foundational understanding of the generative processes, facilitating algorithm prototyping without an explicit reliance on protein structures.

**Strengths:**

1.  **Significant Contribution to Standardization and Reproducibility**: The development of a unified framework and comprehensive benchmark for SE(3)-based protein design is a substantial service to the community. By standardizing training data, implementation details, and evaluation metrics, the authors have created an important resource that enables rigorous, fair, and transparent comparisons among competing methodologies.

2. **Mathematical Decoupling Toolkit**: The introduction of a mathematical decoupling toolkit is a technical contribution. This toolkit provides a clear, principled foundation that articulates the underlying mathematical differences between generative models.

**Weaknesses:**

1. **Need for Enhanced Data Visualization**: The core comparison results presented in Table 1 are difficult to interpret solely in tabular form. Supplementing this data with appropriate visual figures is necessary to effectively communicate the relative performance, elucidate trends across different protein lengths, and enhance the overall clarity and impact of the empirical findings.

**Questions:**

1. Could the authors include a figure representation of the results in Table 1? This visualization should effectively illustrate the performance differences (or similarities) among the various methods, particularly emphasizing how performance metrics trend or vary as a function of protein length or sequence complexity.

2. In Appendix B, the paper draws a conclusion that "SE(3)-based approaches merit greater attention and exploration." This statement is based on comparisons that include models like Proteina, DPLM-2, and MultiFlow, which, as noted, were trained with different architectures, proprietary training data, and potentially distinct task formulations. Could the authors rephrase the conclusion to be more appropriately guarded?

3. There are some typos in the caption of Table 1, please check and correct them.

---

### Official Review · Reviewer_5vXG · 2025-10-26

**Soundness:** 2
**Presentation:** 2
**Contribution:** 1
**Rating:** 2
**Confidence:** 4

**Summary:**

This work introduces Protein-SE(3), a unified benchmark for SE(3)-based protein design models like Genie 1-2, FrameDiff, FoldFlow, and FrameFlow. All baselines are trained and evaluated on same dataset to ensure fair and consistent comparison.

**Strengths:**

The proposed framework facilitates re-training and evaluation of various SE(3)-based baseline models by other researchers.

**Weaknesses:**

## Limited contribution
The proposed benchmark is incomplete, as it only includes SE(3)-based models. Given the diversity of protein design models, models beyond SE(3)-based category should also be included, not left out for future work.

Moreover, since the primary contribution of this work is benchmarking, the experimental evaluation should have been more expensive -- e.g., reporting results across multiple data splits and/or random seeds. These details are not mentioned in the paper.

Finally, the usefulness of the proposed mathematical decoupling toolkit is questionable, as it trains MLP-based flow/diffusion models on simple toy synthetic data, without demonstrating any clear practical application.

## Overstated claims

Section 8 seems to make overstated claims. The authors assert that their decoupling toolkit demonstrates superior efficiency of flow matching in $\mathbb{R}^3$ and $SO(3)$ spaces; however, this conclusion is based solely on toy experiments under limited conditions.

## Unclear motivation

The introduction lacks clarity regarding the motivation and novelty of the benchmark. While the authors state that existing benchmarks “overlook consistent re-training and fair comparison,” no concrete examples or analyses are provided to illustrate how or why current benchmarks fail in these aspects.

### Typos
- Line 84 incomplete sentence
- Line 137 "rotation angel"
- Line 377 "To abstract and abstract"

**Questions:**

Could the authors clarify in what ways previous benchmarks have “overlooked consistent re-training and fair comparison,” and explain how this gap motivated the development of Protein-SE(3)?

---

### Official Review · Reviewer_PZ7B · 2025-10-31

**Soundness:** 3
**Presentation:** 3
**Contribution:** 1
**Rating:** 4
**Confidence:** 4

**Summary:**

This paper describes a codebase that unifies 6 prior frame based protein design generative models. The authors re-train all 6 models on the same dataset and evaluate them on both unconditional generation and motif scaffolding using standard quality, novelty and diversity metrics enabling a fair comparison between the underlying generative methodologies in each case. Toy datasets on translations and rotations are also provided to gain insights into generative processes on those modalities.

**Strengths:**

I think this codebase will be a valuable resource for further research. Due to its fixed training set and evaluation code, it will make it very easy for researchers to add in their own method and have it be immediately comparable with all previous frame-based generative models. The inclusion of the toy datasets for debugging is also nice as this is often a part of model development that researchers usually have to implement themselves.

The metrics used to evaluate seem comprehensive to me, covering sample quality, diversity and novelty in addition to measuring secondary structure content which is a metric that often differentiates methods.

**Weaknesses:**

The motif scaffolding benchmark setup seems quite non-standard with the authors measuring scTM and MotifRMSD values. Previous motif scaffolding bechmarks have used the idea of 'successful designs' that meet certain motif RMSD and scRMSD thresholds and then counting the number of unique and successful designs. I think the authors could benefit from simply integrating a recent motif scaffolding benchmark such as MotifBench.

The plots for secondary structure are nice however it would be ideal to have a quantitative metric with which to compare models rather than having to eyeball different plots. Perhaps the distributional difference to the PDB could be computed.

I don't know why the authors restricted themselves to only frame based models. The codebase could become a training harness for all types of protein generative models which share the same training dataset and evaluation protocol. This seems like a missed opportunity.

Sadly I think that ICLR may not be the correct venue for this work since in terms of originality and significance there is not much new in this work and there are not many new insights. I do however believe the codebase itself is very valuable to the community.

**Questions:**

I wonder if the toy datasets on rotations could be very difficult to learn. It seems to me that a 3D object has been placed in euler angle space in order to create the dataset which seems like quite an unnatural way to generate rotations to learn over and could end up with points near singularities in rotation space.

---

### Official Review · Reviewer_YRLi · 2025-10-31

**Soundness:** 1
**Presentation:** 1
**Contribution:** 1
**Rating:** 2
**Confidence:** 4

**Summary:**

The paper introduces a framework to benchmark protein generative models, including a training set that can be used to retrain the models for fair comparison

**Strengths:**

**Originality**: It is probably not a bad idea to do a fair comparison of these algorithms, with the same training data.

**Clarity**: The figures are well presented

**Weaknesses:**

- There are multiple things about this paper I do not understand. First of all, I do not understand the focus on SE(3) methods. It seems to me like the central claim of the paper is to build a pipeline for evaluating generative models for proteins, including training them all on the same data. But why the focus on SE(3). I see no reason why a method that is not based on SE(3) could not be put through the same pipeline.
- While I think there is value in retraining these models on the same data, to evaluate them, I also do not think it is entirely fair. After all, anyone who has ever developed a model, knows that model choices (architecture, hyperparameters, etc.) are made based on the data at hand. So, retraining these models on a (very small) dataset, that most of them were not designed for, does not necessarily measure their performance. Furthermore, my feeling is that ultimately, what matters is how good the model is at designing proteins, even if that performance is driven by a specific dataset. An example of this are folding models. AlphaFold3 beats any other public reimplementation, because it was trained on more data, even if the implementation are largely the same.
- The training dataset of ~19k proteins is exceedingly small. Some of these models were trained on datasets that were 10x larger. Therefore, performance, particularly in terms of generalization, is never going to be good.
- The models benchmarked are very out of date: There is a FoldFlow2, but the paper seems to use the much older FoldFlow. Likewise with RFDiffusion2. There is also a family of models that do all-atom design (La Proteina, ProtParpadelle, Latent-X) that are not included.
- The quality benchmarks do inverse folding generating 8 sequences with ProteinMPNN, then refolding with ESM-Fold. It seems to me like the number of sequences, and inverse and forward models, should present the user multiple options, and a good benchmark would also ablate over those.
- The authors present DDPM and score-matching as distinct modeling paradigms. However, these are mathematically equivalent in the continuous-time limit, as established by Song et al. (2020). DDPMs can be viewed as a discrete implementation of score-based diffusion, differing mainly in the parameterization of the reverse process (stochastic vs. deterministic formulations)
- There are some very important citations missing, such as Lipman's flow matching paper, which is mind boggling in a paper that talks a lot about flow matching.
- I found section 7 extremely confusing, and out of place. Is it just trying to academically study the differences between DDPM, score-matching and flow matching, unrelated to proteins? If so, there are many available resources that do a very good job at this.

**Questions:**

Aside from the weaknesses highlighted above, there are multiple typos in the text. For example:

- Line 50: "Backend by Pytorch Lightning"
- Line 86: "Based on the and Wasserstein"
- Line 136: "Rotation angel"
- Table 1 caption: "scTM ¿ 0.5"
- Line 271: "flow-matching based methods (FrameFlow and Foldflow) demonstrates"
- Line 272: "Novelty is also an essential metric to evaluates"
- Line 276" "shows a decline trend"
- Line 377: "To abstract and abstract"

Terms like Wasserstein, FoldFlow and RFDiffusion also do not consistently use uppercase

---

### Note · Authors · 2025-11-12

I have read and agree with the venue's withdrawal policy on behalf of myself and my co-authors.